# 3D Numerical Simulation of a Z Gate Layout MOSFET for Radiation Tolerance

**DOI:** 10.3390/mi9120659

**Published:** 2018-12-14

**Authors:** Ying Wang, Chan Shan, Wei Piao, Xing-ji Li, Jian-qun Yang, Fei Cao, Cheng-hao Yu

**Affiliations:** 1Key Laboratory of RF Circuits and Systems, Ministry of Education, Hangzhou Dianzi University, Hangzhou 310018, China; luoxin07@hrbeu.edu.cn (W.P.); caofei@hdu.edu.cn (F.C.); yuchenghao@hdu.edu.cn (C.-h.Y.); 2College of Information Engineering, Jimei University, Xiamen 361021, China; shanchan@jmu.edu.cn; 3National Key Laboratory of Materials Behavior and Evaluation Technology in Space Environment, Harbin Institute of Technology, Harbin 150080, China; lxj0218@hit.edu.cn (X.-j.L.); yangjianqun@hit.edu.cn (J.-q.Y.)

**Keywords:** bulk NMOS devices, radiation hardened by design (RHBD), total ionizing dose (TID), Sentaurus TCAD, layout

## Abstract

In this paper, for the first time, an n-channel metal-oxide-semiconductor field-effect transistor (NMOSFET) layout with a Z gate and an improved total ionizing dose (TID) tolerance is proposed. The novel layout can be radiation-hardened with a fixed charge density at the shallow trench isolation (STI) of 3.5 × 10^12^ cm^−2^. Moreover, it has the advantages of a small footprint, no limitation in *W*/*L* design, and a small gate capacitance compared with the enclosed gate layout. Beside the Z gate layout, a non-radiation-hardened single gate layout and a radiation-hardened enclosed gate layout are simulated using the Sentaurus 3D technology computer-aided design (TCAD) software. First, the transfer characteristics curves (*I*_d_-*V*_g_) curves of the three layouts are compared to verify the radiation tolerance characteristic of the Z gate layout; then, the threshold voltage and the leakage current of the three layouts are extracted to compare their TID responses. Lastly, the threshold voltage shift and the leakage current increment at different radiation doses for the three layouts are presented and analyzed.

## 1. Introduction

The total ionizing dose (TID) effect is one of the mechanisms that causes radiation-induced anomalies in semiconductor devices. The TID mechanism induces the generation of trapped charges in the dielectrics and interface states along the Si/SiO_2_ interfaces, causing degradation of a transistor’s performance [1,2,3,4]. Due to the downscaling, the net-charge trapping in oxides with a thickness of less than 10 nm is modest [5,6,7,8,9]. Since the thickness of the gate oxide of the simulated transistors is 2 nm, in this work, the net-charge trapping in the oxides is negligible. Therefore, the effects on thick oxides, such as the shallow trench isolation (STI), dominate the TID response of metal-oxide-semiconductor field-effect transistors (MOSFETs) [10]. Moreover, the charge trapped in the spacer oxide or at its interface modifies the parasitic series’ resistance, reducing the drive current [11].

In a conventional non-radiation-hardened single gate layout, the STI’s parasitic conduction path (the red arrow in Figure 1a) induced by the TID effect, which is visible only in an n-MOSFET, occurs along the sidewall oxide between the source and the drain, and leads to an increase in the drain current as the radiation dose increases [1]. A widely studied layout with radiation hardness, called the enclosed gate layout [12,13,14], which requires tradeoffs in application [14,15], is presented in Figure 1b. For instance, a very small width over length ratio (*W*/*L*) is not realistic for an Enclosed Layout Transistor (ELT), for which the minimum achievable *W*/*L* is 2.26 [15], which is a significant concern in analog circuits [16]. Moreover, a larger gate capacitance will cause a longer time delay, which is not favorable for digital circuits. A large footprint is another disadvantage of the enclosed gate layout. In circuit design, the area penalty induced by design has been the main drawback [17].

In order to eliminate the parasitic path and overcome the disadvantages of an enclosed gate layout, for the first time, an n-MOSFET layout with a Z gate is proposed. Moreover, the proposed Z gate layout is applicable to more complicated structures, such as fin-field-effect transistors (FinFETs), tunnel-field-effect transistors (TFETs), and nanowires [18,19,20,21,22]. In this paper, devices with the proposed Z gate layout achieve total-dose hardness by eliminating these edges, but at the expense of fabrication feasibility due to the asymmetric active area design, as shown in Figure 1c. First, the effectiveness of the proposed layout to eliminate the leakage current is demonstrated by *I*_d_-*V*_g_ curves. Then, the total shift of the threshold voltage and the variation of the leakage current, before and after the radiation is applied, are calculated for the single gate layout, the enclosed gate layout, and the Z gate layout, respectively. Further, the three simulated layouts are compared with respect to the threshold voltage shift and the leakage current increase as a function of the fixed charge density. Comparing the static characteristics of the different transistor layouts, it is found that the Z gate layout exhibits the best TID response compared with the conventional layouts and ELTs.

## 2. Device Structure and Simulation

The Z gate layout achieves the radiation hardness by introducing two short extra gates that separate the active area and the isolation oxides. It should be noted that the precise, effective *W*/*L* ratio model of the proposed layout is not available at present; so, the channel width of the Z gate layout in this work is defined as shown in Figure 1c. A report [15] proposed an effective *W*/*L* model of an enclosed gate layout, and concluded that the only way to obtain a low aspect ratio is to increase the L value. In the rectangular shape of an enclosed gate layout, the minimum *W*/*L* achievable is 2.26, and is almost reached with *L* = 7 um [15], which implies a considerable waste of area and a large capacitance issue. Although a precise *W*/*L* model of the Z gate layout is not available at present, the drain current level of the Z gate layout, when compared with the drain current of a single gate layout with the same *W*, *L*, and the overdrive voltage (*V*_gt_, *V*_gt_ = *V*_gs_ − *V*_th_), is nearly the same. It assumes that a Z gate layout does not need to increase the *L* value that high to achieve the same effective *W*/*L* with a single gate layout, and, thus, has a smaller footprint and gate capacitance.

In the simulation, the main parameters were kept the same for all three layouts. The lateral spacers were formed by a layer of SiO_2_ and a thick layer of Si_3_N_4_, and the STI was inserted using the SiO_2_. Because the enclosed gate layout was not able to achieve a small *W*/*L* at *L* = 0.12 μm, the values of *R*_1_ and *R*_2_ were 0.15 μm and 0.27 μm, respectively, and the effective *W*/*L* was calculated by the formula given in [14], and it was equal to 13.6. The main parameters of the transistors in the simulation are listed in Table 1.

The TID effect on the MOSFET was modeled by adopting the fixed-charge insulator model provided by the sentaurus technology computer-aided design (TCAD) software, which can be used to set a fixed charge density between the STI and the active region [23]. All simulations were performed using a hydrodynamic model with high-field saturation and mobility degradation models that included doping dependence and carrier–carrier scattering. We simulated the effects of the total radiation dose by increasing the fixed charge density on the sidewall oxide [24]. It should be noted that this work is focused only on the effects of fixed charges; so, the interface states were neglected for the reasons below. When a complementary metal-oxide-semiconductor (CMOS) device is exposed to radiation, hole trapping results in fixed charges and interface states in the thick oxides. According to a report on the radiation-induced fixed charge density and interface state density in MOS capacitors [25], the radiation-induced flat band voltage is predominantly shifted by the fixed charges. The effect of the interface states is minor. Therefore, in this simulation, the interface states were neglected, and only the fixed charge density was modified to reflect the total ionizing dose effect [24]. Moreover, the effects of interface traps were left out of the simulations due to a lack of empirical information about several parameters of interface traps, such as trap energy and density and the capture cross-section, which are necessary for accurate simulations [26]. In addition, we can see from the literature [26] that the tendencies of Δ*V*_th_ and ΔSS extracted from the simulation results are in good agreement with those from the experimental data of 5 Mrad. Through the three-dimensional (3D) simulation results, they confirm that, for sub-100 nm gate-all-around metal-oxide-semiconductor field-effect transistors (GAA MOSFETs), the fixed charges in the gate spacer predominantly determine Δ*V*_th_ and Δ*SS*, i.e., the TID effect. Note that interface traps were not taken in the simulation in this paper. Although that may result in some disagreement in the current levels as obtained with the experimental counterparts, this case does not have much impact on our findings, because the focus of this paper is not on the exact values of currents but on the general trends and relative results of Z gate, enclosed gate, and single gate layouts due to the TID effect.

## 3. Results and Discussion

### 3.1. The I_d_-V_g_ Simulation Results

In order to verify that the Z gate layout is able to work well in a non-radiation environment, we simulated the *I*_d_-*V*_g_ curves of the Z gate layout, the enclosed gate layout, and the single gate layout at the fixed charge density of 3 × 10^10^ cm^−2^ to model the non-radiation scenario. The following simulation results focus on the analysis of the radiation tolerance characteristics of the proposed layout. The degradation of devices is mainly characterized by the threshold-voltage shift and the off-state leakage current [27]. As we know, *I*_DSS_ is the maximum current that flows through a FET transistor, which is when the gate voltage (VG) supplied to the FET is 0 V. Additionally, it is only valid when the FET transistor is a junction field-effect transistor (JFET) or depletion MOSFET. However, as the proposed Z gate layout transistor is an enhanced MOSFET, we think that the parameters of *I*_DSS_ and the *I*_DSS_/*I*off ratio are unnecessary to investigate. The threshold-voltage is determined by the linear extrapolation method in the linear region; thus, the simulation was performed with a very small *V*_DS_, i.e., 20 mV, 50 mV, and 100 mV [24,27,28]. In this paper, *V*_DS_ was taken as 20 mV. Moreover, the TID effect will be more serious when *V*_DS_ reaches *V*_DD_ [27]. This can be mainly attributed to the fact that more trapped charges at the STI/body interface will sufficiently reduce the potential barrier and result in a larger leakage current at a high drain voltage. In addition, the use of a 20 mV drain bias gives the best results for the Z gate layout in comparison to the alternatives (results not shown). Thus, in this paper, the three layout types are simulated at the drain bias of 20 mV, which sweeps the gate bias from 0 V to 1.5 V.

The simulation results of the *I*_d_-*V*_g_ curves of the single gate layout are shown in Figure 2a, where it can be seen that the leakage current significantly increases as the fixed charge density increases, and that the on-current increases slightly as the fixed charge density increases. The simulation results of the *I*_d_-*V*_g_ curves of the enclosed gate layout are shown in Figure 2b, where the *I*_d_-*V*_g_ curves almost overlap with each other, demonstrating a small impact of the TID effect on the enclosed gate layout.

The simulation results of the *I*_d_-*V*_g_ curves of the Z gate layout are shown in Figure 2c, wherein it can be seen that the leakage current increases slightly as the fixed charge density increases. The radiation tolerance characteristic of the Z gate layout was verified by comparison with that of the single gate layout. The curves of the Z gate layout are similar to those of the enclosed gate layout; namely, the leakage current increased very little as the fixed charge density increased, demonstrating that the Z gate layout was radiation tolerant at the fixed charge densities at the STI of 3.5 × 10^12^ cm^−2^, the same as the enclosed gate layout.

### 3.2. Comparison of Key Transistor Performance Parameters

To compare the TID response of the transistors fairly, the threshold voltage and leakage current parameters were extracted at the fixed charge density of 3 × 10^10^ cm^−2^ and 3.5 × 10^12^ cm^−2^ to model the pre- and post-radiation scenarios, respectively. The results of the threshold voltage and leakage current are listed in Table 2 and Table 3, respectively.

In Table 2, the threshold voltage of the non-radiation-hardened single gate layout at pre- and post-radiation is 363 mV and 138 mV, respectively, and the total shift is 225.56 mV; for the other two radiation-hardened layouts, the total shift is below 30 mV. Thus, regarding the shift value in descending order, the order of three layout types is the single gate layout, the Z gate layout, and the enclosed gate layout.

In Table 3, the leakage current of the single gate layout pre- and post-radiation is 0.458 nA and 3.44 μA, respectively, and the total shift is approximately 3.44 μA. The order of magnitude of the leakage current increase of the other two radiation-hardened layouts was about 1×10^−9^ A compared to the enclosed gate layout. Regarding the increment value in descending order, the order of the three layout types is the single gate layout, the Z gate layout, and the enclosed gate layout.

The above-presented comparison of the three layout types regarding the two parameters demonstrates that the enclosed gate layout achieved the best radiation-hardness performance, and the Z gate layout was more effective in mitigating the TID effect on the transistor than the single gate layout.

The threshold voltage shift and the leakage current of the single gate layout, the Z gate layout, and the enclosed gate layout at different charge densities are depicted in Figure 3 and Figure 4, respectively. In Figure 3, it can be seen that the threshold voltage shift of the single gate layout changed non-linearly with the fixed charge density. The shift increased rapidly at a low charge density, and then slowly decreased after reaching the peak value at the fixed charge density of about 2×10^12^ cm^−2^. As can be clearly seen in the inner figure in Figure 3, the enclosed gate layout’s shift was kept very small, and the Z gate layout’s shift was similar to that of the single gate layout. It is shown that the shift value of the single gate layout at different charge densities was much larger than that of the other two layouts. The largest shift value of the single gate layout, the Z gate layout, and the enclosed gate layout was 37 mV, 6 mV, and 0.09 × 10^−3^ mV, respectively. In the inner figure in Figure 3, it can be seen that the enclosed gate layout’s shift was smaller than that of the Z gate layout. The enclosed gate layout achieved great performance regarding the radiation hardness; however, the enclosed gate layout comes with disadvantages that cannot be ignored, hindering its application to certain circuits. In such situations, the Z gate layout is a better solution.

The difference between the three layouts regarding the leakage current was even more obvious. As shown in Figure 4, the leakage current of the single gate layout increased rapidly at a low charge density, and then slowly decreased after reaching the peak value at the fixed charge density of about 2 × 10^12^ cm^−2^, showing a similar trend as the other two layouts (the trend of the enclosed gate layout is not shown in Figure 4), but at a different order of magnitude. The order of magnitude of the leakage current increase of the single gate layout was about 1 × 10^−7^ A, and for the other two layouts, it was about 1 × 10^−10^ A. Thus, it is shown that, compared with the single gate layout, the leakage current increase of the other two layouts was very small, demonstrating the great radiation-hardened characteristic of these two layouts. As can be seen in the inner figure in Figure 4, the leakage current increase of the enclosed gate layout was still relatively small, showing the best radiation tolerance among the three layouts. The largest leakage current increase of the single gate layout, the enclosed gate layout, and the Z gate layout was 0.66 μA, 1.06 nA, and 0.8 × 10^−3^ nA, respectively. The reduction in the leakage current of the enclosed gate layout and of the Z gate layout compared with the single gate layout was 0.660 μA and 0.659 μA, respectively. According to the results, the Z gate layout performed differently from the enclosed gate layout; however, the Z gate layout was still as effective as the enclosed gate layout regarding the leakage current reduction. Consequently, the Z gate is a better solution at the fixed charge density of 3.5 × 10^12^ cm^−2^. In addition, the radiation effects will deteriorate even more as the channel length shrinks due to short-channel effects, which are attributed to the positive charge trapped at the STI/body interface by the radiation [27,28]. This is a serious problem for a highly scaled device operating in an irradiated environment.

## 4. Conclusions

A novel n-MOSFET layout with a Z gate was proposed and analyzed using the Sentaurus 3D TCAD software. By comparing the proposed layout with the single gate layout and the enclosed gate layout with respect to the threshold voltage and the leakage current, the radiation-hardened characteristic of the Z gate layout was verified. Besides this, the proposed layout effectively reduces the impact of the TID effect on the transistor’s performance compared with the single gate layout; also, the Z gate layout overcomes the drawbacks of the enclosed gate layout, such as a large footprint, a limitation in the *W*/*L*’s design, and a large capacitance. Thus, the Z gate is a better solution at the fixed charge density of 3.5 × 10^12^ cm^−2^.

## Figures and Tables

**Figure 1 micromachines-09-00659-f001:**
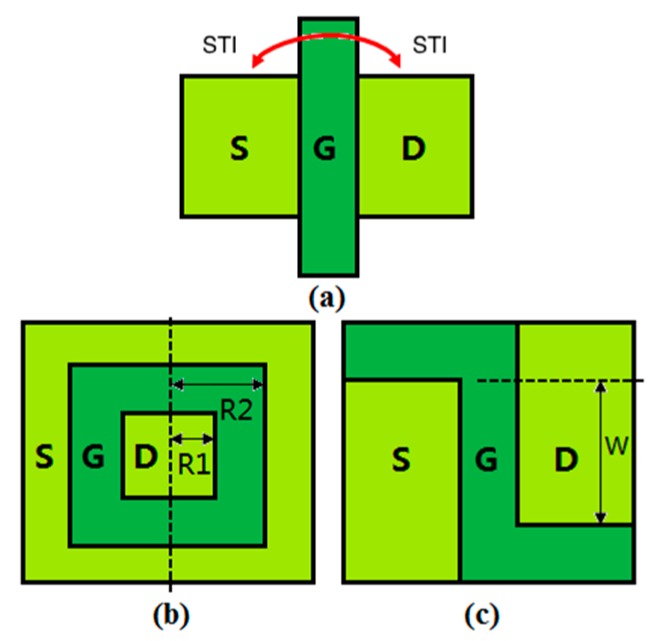
Schematic structures of (**a**) the conventional layout, (**b**) the H gate layout, and (**c**) the proposed layout. STI, shallow trench isolation.

**Figure 2 micromachines-09-00659-f002:**
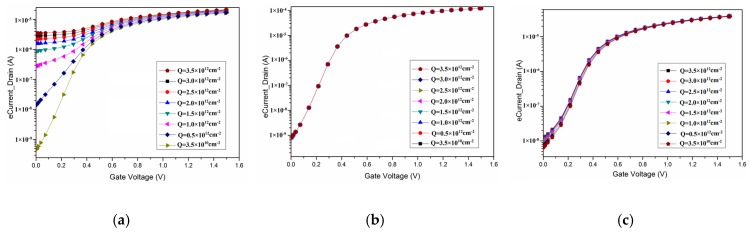
The simulation results of the *I*_d_-*V*_g_ curve of (**a**) the single gate layout, (**b**) the enclosed gate layout, and (**c**) the Z gate layout.

**Figure 3 micromachines-09-00659-f003:**
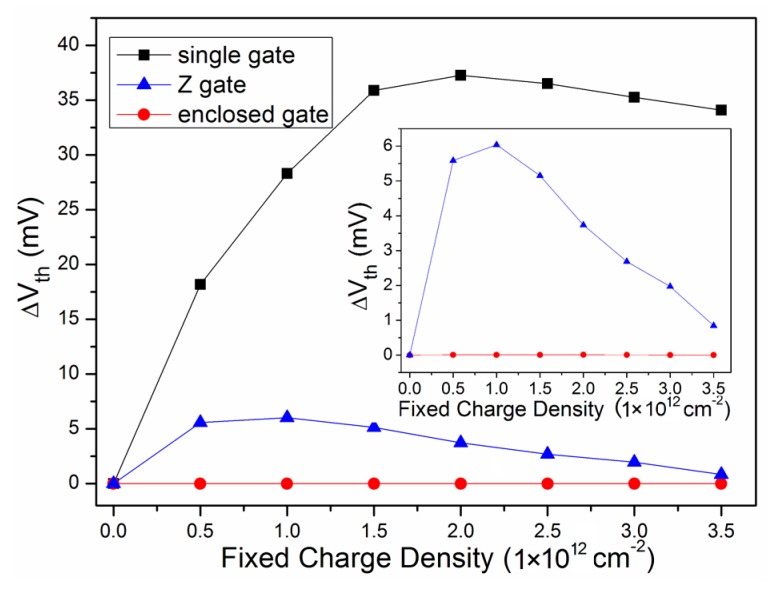
The threshold voltage shift of the three layouts at different charge densities.

**Figure 4 micromachines-09-00659-f004:**
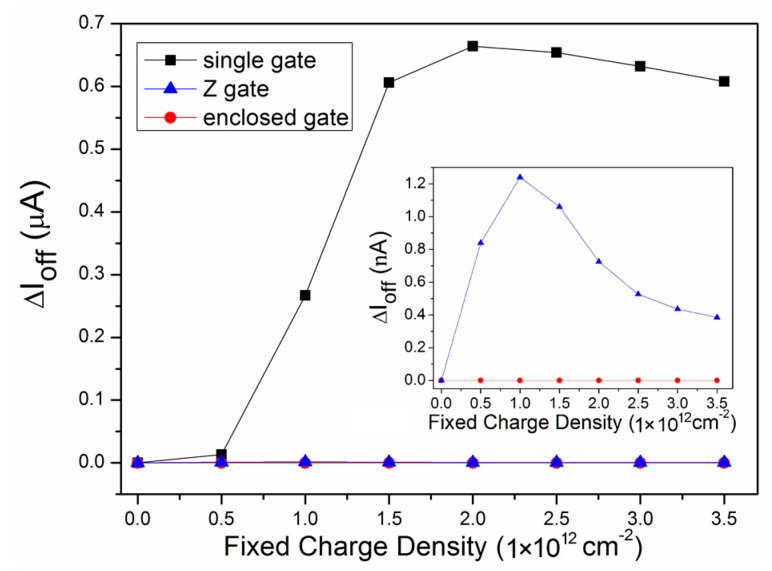
The leakage current increment of the three layouts at different charge densities.

**Table 1 micromachines-09-00659-t001:** The parameters that were used for the device’s simulation.

Parameter	Value
Length of channel	0.12 μm
Width of channel	0.21 μm
Thickness of n-type poly gate	100 nm
Thickness of gate oxide	2 nm
Doping of source/drain region	1.0 × 10^19^ cm^−3^
Depth of source/drain region	100 nm
Doping of p-type substrate	4.0 × 10^17^ cm^−3^

**Table 2 micromachines-09-00659-t002:** V_th_ in the pre- and post-radiation scenarios.

Layout	V_th_-pre (mV)	V_th_-post (mV)	ΔV_th_ (mV)
single gate	363	138	226
enclosed gate	374	374	<1
Z gate	354	329	25

**Table 3 micromachines-09-00659-t003:** I_off_ in the pre- and post-radiation scenarios.

Layout	I_off_-pre (A)	I_off_-post (A)	Increment
single gate	4.58 × 10^−10^	3.44 × 10^−6^	3.44 μA
enclosed gate	7.91 × 10^−10^	7.95 × 10^−10^	0.004 nA
Z gate	6.46 × 10^−9^	1.17 × 10^−8^	5.24 nA

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
