# Peer review of "3D Numerical Simulation of a Z Gate Layout MOSFET for Radiation Tolerance"

_micromachines, 2018, doi:10.3390/mi9120659_

Round 1
Reviewer 1 Report
In this study, an n-MOSFET layout with the Z gate with improved total ionizing dose (TID) tolerance is proposed and three simulated layouts are compared in the aspect of the threshold voltage shift and the leakage current increase as a function of the fixed charge density. This paper is of acceptable scientific quality and can be accepted after major revisions. The following comments, however, should be addressed before publication.
Comments:
1. Page 1, line 28-35 (Introduction section). Some recent reports should be cited instead of outdated or earlier papers.
2. Page 2, Figure 1. I recommend that the cross-sectional structures of three simulated layouts may include in this paper for improving the readability.
3. Page 2, line 77. The effective W/L was calculated by the formula given in [8], and it was equal to 13.6. Is the value of W/L an optimal parameter? Please explain in detail.
4. Table 1. The parameters used for devices simulation do not include active channel layer. Why?
5. Page 3, line 92-93. The three layout types were simulated at the drain bias of 20 mV sweeping the gate bias from 0 V to 1.5 V. I strongly recommend that the gate bias should sweep from negative voltage to positive voltage for understanding the contact behavior in Figure 2 and off current state in Figure 3 and 4.
6. Characters are too small in several figures.
Author Response
Response to Reviewer 1 Comments
In this study, an n-MOSFET layout with the Z gate with improved total ionizing dose (TID) tolerance is proposed and three simulated layouts are compared in the aspect of the threshold voltage shift and the leakage current increase as a function of the fixed charge density. This paper is of acceptable scientific quality and can be accepted after major revisions. The following comments, however, should be addressed before publication.
Point 1: Page 1, line 28-35 (Introduction section). Some recent reports should be cited instead of outdated or earlier papers.
Response 1:. Thank you for your suggestion. The manuscript has been modified based on your advice.
The total ionizing dose (TID) effect is one of the mechanisms which causes the radiation-induced anomalies in the semiconductor devices. The TID mechanism induces the generation of trapped charges in the dielectrics and interface states along the Si/SiO2 interfaces, causing degradation of transistors performance [1-4]. Due to the down-scaling, the net charge trapping in the oxides with the thickness of less than 10 nm is modest [5-9].
[1] Schwank, J.R.; Shaneyfelt, M.R.; Fleetwood, D.M.; Felix, J.A.; Dodd, P.E.; Paillet, P.; Ferlet-Cavrois, V. Radiation Effects in MOS Oxides. IEEE Transactions on Nuclear Science. 2008, 55(4), 1833-1853. DOI: 10.1109/TNS.2008.2001040.
[2] Borghello, G.; Faccio, F.; Lerario, E.; Michelis, S.; Kulis, S.; Fleetwood, D. M.; Bonaldo, S. Dose rate sensitivity of 65 nm MOSFETs exposed to ultra-high doses. IEEE Transactions on Nuclear Science. 2018, 65(8), 1482-1487. DOI: 10.1109/TNS.2018.2828142.
[3] Saremi, M.; Privat, A.; Barnaby, H. J.; Clark, L. T. Physically Based Predictive Model for Single Event Transients in CMOS Gates. IEEE Transaction on Electron Devices. 2016, 63(6), 2248-2254. DOI: 10.1109/ted.2016.2547423.
[4] Narasimham, B.; Bhuva, B. L.; Schrimpf, R. D.; Massengill, L. W.; Gadlage, M. J.; Amusan, O. A.; Benedetto, J. M. Characterization of digital single event transient pulse-widths in 130-nm and 90-nm CMOS technologies. IEEE Transactions on Nuclear Science. 2007, 54(6), 2506-2511. DOI: 10.1109/TNS.2007.910125.
[5] Barnaby, H. J. Total-ionizing-dose effects in modern CMOS technologies. IEEE Transactions on Nuclear Science. 2006, 53(6), 3103–3121. DOI: 10.1109/TNS.2006.885952.
[6] Faccio, F; Cervelli, G. Radiation-induced edge effects in deep submicron CMOS transistors. IEEE Transactions on Nuclear Science. 2005, 52(6), 2413–2420. DOI: 10.1109/TNS.2005.860698.
[7] Zhang, C. M.; Jazaeri, F.; Borghello, G.; Faccio, F.; Mattiazzo, S.; Baschirotto, A.; Enz, C. Characterization and Modeling of Gigarad-TID-induced Drain Leakage Current of 28-nm Bulk MOSFETs. IEEE Transactions on Nuclear Science. 2018, DOI: 10.1109/TNS.2018.2878105.
[8] Sajid, M.; Chechenin, N. G.; Torres, F. S.; Hanif, M. N.; Gulzari, U. A.; Arslan, S.; Khan, E. U. Analysis of Total Ionizing Dose effects for highly scaled CMOS devices in Low Earth Orbit. Nuclear Instruments and Methods in Physics Research Section B: Beam Interactions with Materials and Atoms. 2018, 428, 30-37. DOI: 10.1016/j.nimb.2018.05.014.
[9] Fleetwood, D. M. Evolution of Total Ionizing Dose Effects in MOS Devices with Moore’s Law Scaling. IEEE Transactions on Nuclear Science. 2018, 65(8), 1465-1481. DOI: 10.1109/TNS.2017.2786140.
Point 2: Page 2, Figure 1. I recommend that the cross-sectional structures of three simulated layouts may include in this paper for improving the readability.
Response 2: Thank you for your suggestion. Figure.1 shows the top view of the three simulated layouts, which illustrates all the differences between the three structures.
Point 3: Page 2, line 77. The effective W/L was calculated by the formula given in [8], and it was equal to 13.6. Is the value of W/L an optimal parameter? Please explain in detail.
Response 3: Thank you for your suggestion. The value of W/L is not an optimal parameter here, it’s just a common parameter we use to study.
Point 4: Table 1. The parameters used for devices simulation do not include active channel layer. Why?
Response 4: Thank you for your suggestion. The simulated devices are conventional bulk-Si transistor, and the channel length is 0.12μm, which is beyond 100nm. Thus, the active channel layer is not a significant parameter and we can select it empirically. Therefore, Table 1 does not include ctive channel layer.
Point 5: Page 3, line 92-93. The three layout types were simulated at the drain bias of 20 mV sweeping the gate bias from 0 V to 1.5 V. I strongly recommend that the gate bias should sweep from negative voltage to positive voltage for understanding the contact behavior in Figure 2 and off current state in Figure 3 and 4.
Response 5: Thank you for your suggestion. Since the threshold voltages (VTH) of the three devices are all beyond 0V in the simulation, the gate overdrive voltage VGT (VGT=VGS-VTH) sweeps from negative voltage when the gate bias starts from 0V.
Point 6: Characters are too small in several figures.
Response 6: Thank you for your suggestion. The figures have been modified based on your advice.
Figure 3. Threshold voltage shift of three layouts at different charge density.
Figure 4. Leakage current increment of three layouts at different charge density.

Reviewer 2 Report
Dear Editor and Authors,
I have read the paper entitled "3-D Numerical Simulation of Z Gate Layout MOSFET for Radiation Tolerance", which deals with a very interesting and relevant topic that is attractive for many readers interested in novel/modern MOS devices. It also has been a welcome change to finally have paper written in fluent and proper English which was a pleasure to read. Moreover, content-wise, the paper is of very good quality. I would have recommended it for immediate publication were it not for some corrections, inclduing some which -after much pondering- I considered were of great importance, hence my final recommendation that MAJOR corrections are needed.
The corrections/additions the paper needs are listed below, as follows:
MAJOR:
- In section 3.2 & 3.3, I suggest a few changes & additions. Firstly, change the title of the section into "Comparison of key transistor performance parameters", and merge together sections 3.2 & 3.3. Second, I suggest the Authors should also add in their analysis/comparison, the data for transconductance, IDSS (@ the Max. biasing using by the Authors in their simulations or -much better!- at a much more realistic value, see my next comment) and the IDSS/Ioff ratio. (Or, if the authors prefer it, to add new small sections for each parameter). This would address the rather short length of the paper (in my opinion).
Also, the drain bias of only 20 mV seems extremely small. The Authors should justify very strongly why the took this very small value in consideration, as to me it seems rather unrealistic. In practice it may very well be that at least some of the transistors could be biased at VDD, hence the Authors should perform simulations using the battery voltage value typically used in most applications of IC using the technology node assumed by the Authors for their transistors size.
- Since the Authors discuss devices for improved radiation-hardened ICs, could it be that the ionization radiation may also have other effects, e.g. modify the number of surface states at the oxide-channel interface? Should these effects be considered in simulations as well, and if no, why?
- I think the main major weak point of the paper is that it is purely theoretical. It would be excellent, and greatly strengthen the paper, if the Authors could add at least a few practical measurements of some realized devices, even if to verify the simulated results just for a single charge density value.
MINOR:
- Insert blanks between numbers and the measurement units, e.g. in Abstract replace "3.5×10^12cm^-2" with ""3.5×10^12 cm^-2".
- At p.1, line 31, replace "scaling" with "down-scaling", and in 1st top line of Table 3 replace "Increament" with "Incrament".
Once these corrections/additions are done, I believe that the Authors can forward the revised paper directly to the Editor for faster publication.
With best wishes,
The reviewer
Author Response
Response to Reviewer 2 Comments
I have read the paper entitled "3-D Numerical Simulation of Z Gate Layout MOSFET for Radiation Tolerance", which deals with a very interesting and relevant topic that is attractive for many readers interested in novel/modern MOS devices. It also has been a welcome change to finally have paper written in fluent and proper English which was a pleasure to read. Moreover, content-wise, the paper is of very good quality. I would have recommended it for immediate publication were it not for some corrections, inclduing some which -after much pondering- I considered were of great importance, hence my final recommendation that MAJOR corrections are needed.
The corrections/additions the paper needs are listed below, as follows:
Point 1. MAJOR:
- In section 3.2 & 3.3, I suggest a few changes & additions. Firstly, change the title of the section into "Comparison of key transistor performance parameters", and merge together sections 3.2 & 3.3. Second, I suggest the Authors should also add in their analysis/comparison, the data for transconductance, IDSS (@ the Max. biasing using by the Authors in their simulations or -much better!- at a much more realistic value, see my next comment) and the IDSS/Ioff ratio. (Or, if the authors prefer it, to add new small sections for each parameter). This would address the rather short length of the paper (in my opinion).
Also, the drain bias of only 20 mV seems extremely small. The Authors should justify very strongly why the took this very small value in consideration, as to me it seems rather unrealistic. In practice it may very well be that at least some of the transistors could be biased at VDD, hence the Authors should perform simulations using the battery voltage value typically used in most applications of IC using the technology node assumed by the Authors for their transistors size.
- Since the Authors discuss devices for improved radiation-hardened ICs, could it be that the ionization radiation may also have other effects, e.g. modify the number of surface states at the oxide-channel interface? Should these effects be considered in simulations as well, and if no, why?
- I think the main major weak point of the paper is that it is purely theoretical. It would be excellent, and greatly strengthen the paper, if the Authors could add at least a few practical measurements of some realized devices, even if to verify the simulated results just for a single charge density value.
Response 1: Thank you for your suggestion. Section 3.2 & 3.3 have been modified based on your advice.
In order to eliminate the parasitic path and overcome the disadvantages of an enclosed gate layout, the novel n-MOSFET layout with the Z gate is proposed. In this paper, the proposed Z gate layout devices achieve total-dose hardness by eliminating these edges. Thus, we only focus on the threshold voltage shift and leakage current variation.
In this paper, we took the drain bias as 20mV on the basis of the ref [1]. “The simulation was performed with a drain-to-source voltage of 0.05 V” in ref [1].
When the total ionizing dose effect was simulated in the designed structures, the fixed charge density was adjusted on the sidewall oxide. When a CMOS device is exposed to radiation, hole trapping results in fixed charges and interface states in the thick oxides. According to a report on the radiation-induced fixed charge density and interface state density in MOS capacitors [2], the radiation-induced flat band voltage is dominantly shifted by the fixed charges. The effect of the interface states is minor. Therefore, in this simulation, the interface states were neglected and only the fixed charge density was modified to reflect the total ionizing dose effect [1].
[1] Lee, M.S.; Lee, H.C. Dummy Gate-Assisted n-MOSFET Layout for a Radiation-Tolerant Integrated Circuit. IEEE Transactions on Nuclear Science. 2013, 60(4), 3084-3091. DOI: 10.1109/TNS.2013.2268390.
[2] Fernandez-Martinex, P.; Cortes, I.; Hidalgo, S.; Flores, D.; Palomo, F. R. Simulation of total ionizing dose in MOS capacitors. Proceedings of the 8th Spanish Conference on Electron Devices. 2011, DOI: 10.1109/sced.2011.5744251.
Due to the limitation of our present experimental conditions, we are unable to provide experimental results. We feel sorry for this. What we have done will advance our further work, and provide a good theoretical foundation and guidance for experimental work. If we have the experimental conditions, the further research work will be carried out to verify the theoretical and simulation results, and the corresponding experimental results will send to this journal.
Point 2. MINOR:
- Insert blanks between numbers and the measurement units, e.g. in Abstract replace "3.5×10^300px^-2" with ""3.5×10^12 cm^-2".
- At p.1, line 31, replace "scaling" with "down-scaling", and in 1st top line of Table 3 replace "Increament" with "Increment".
Response 2: Thank you for your suggestion. The manuscript has been modified based on your advice.

Reviewer 3 Report
Please see the attachment!

Author Response
Response to Reviewer 3 Comments
This manuscript proposes a Z-gate MOSFET layout with improved total ionizing dose (TID) tolerance. This layout has advantages of small footprint, no limitation of W/L design, and small gate capacitance compared with the enclosed gate layout. The layout is compared with the non- radiation-hardened single gate layout and the radiation-hardened enclosed gate layout by using 3D TCAD Sentaurus tool. I think the idea and results are interesting but to improve the quality of the manuscript, I have the following comments:
Point 1: At the beginning of the introduction section, it is better to enrich the introductory materials with previous publications which focused on radiation effects in CMOS technologies such as the one presented below:
a. Dose rate sensitivity of 65 nm MOSFETs exposed to ultra-high doses, IEEE Transactions on Nuclear Science, vol. 65, issue 8, pp 1482-1487, 2018.
b. Physically Based Predictive Model for Single Event Transients in CMOS Gates, IEEE Transaction on Electron Devices, vol. 63, issue 6, pp 2248-2254, 2016.
c. Characterization of digital single event transient pulse-widths in 130-nm and 90-nm CMOS technologies, IEEE Transactions on Nuclear Science, vol. 54, issue 6, pp 2506-2511, 2007.
Response 1: Thank you for your suggestion. The manuscript has been modified based on your advice.
The total ionizing dose (TID) effect is one of the mechanisms which causes the radiation-induced anomalies in the semiconductor devices. The TID mechanism induces the generation of trapped charges in the dielectrics and interface states along the Si/SiO2 interfaces, causing degradation of transistors performance [1-4].
[1] Schwank, J.R.; Shaneyfelt, M.R.; Fleetwood, D.M.; Felix, J.A.; Dodd, P.E.; Paillet, P.; Ferlet-Cavrois, V. Radiation Effects in MOS Oxides. IEEE Transactions on Nuclear Science. 2008, 55(4), 1833-1853. DOI: 10.1109/TNS.2008.2001040.
[2] Borghello, G.; Faccio, F.; Lerario, E.; Michelis, S.; Kulis, S.; Fleetwood, D. M.; Bonaldo, S. Dose rate sensitivity of 65 nm MOSFETs exposed to ultra-high doses. IEEE Transactions on Nuclear Science. 2018, 65(8), 1482-1487. DOI: 10.1109/TNS.2018.2828142.
[3] Saremi, M.; Privat, A.; Barnaby, H. J.; Clark, L. T. Physically Based Predictive Model for Single Event Transients in CMOS Gates. IEEE Transaction on Electron Devices. 2016, 63(6), 2248-2254. DOI: 10.1109/ted.2016.2547423.
[4] Narasimham, B.; Bhuva, B. L.; Schrimpf, R. D.; Massengill, L. W.; Gadlage, M. J.; Amusan, O. A.; Benedetto, J. M. Characterization of digital single event transient pulse-widths in 130-nm and 90-nm CMOS technologies. IEEE Transactions on Nuclear Science. 2007, 54(6), 2506-2511. DOI: 10.1109/TNS.2007.910125.
Point 2: Is the proposed z-gate layout applicable to more complicated structures like FinFETs, TFETs, and nanowires? If so, the authors can briefly mention these structures in the introduction and explain their method (z-gate layout) is applicable to such structures while referring to such publications:
a. A Novel PNPN-Like Z-Shaped Tunnel Field-Effect Transistor With Improved Ambipolar Behavior and RF Performance, IEEE Transactions on Electron Devices, vol. 64, issue 11, pp 4752-4758, 2017.
b. Gate fringe-induced barrier lowering in underlap FinFET structures and its optimization, IEEE Electron Device Letters, vol. 29, issue 1, pp 128-130, 2008.
c. Double-Gate Tunnel FET With High-k Gate Dielectric, IEEE Transactions on Electron Devices, vol. 54, issue 7, pp 1725-1733, 2007.
d. A Resonant Tunneling Nanowire Field Effect Transistor with Physical Contractions: A Negative Differential Resistance Device for Low Power Very Large Scale Integration Applications, Journal of Electronic Materials, vol. 47, issue 2, pp 1091-1098, 2018.
e. Theory of the Junctionless Nanowire FET, IEEE Transactions on Electron Devices, vol. 58, issue 9, pp 2903-2910, 2011.
Response 2: Thank you for your suggestion. The manuscript has been modified based on your advice.
Moreover, proposed Z gate layout is applicable to more complicated structures like FinFETs, TFETs, and nanowires [18-22].
[18] Imenabadi, R. M.; Saremi, M.; Vandenberghe, W. G. A Novel PNPN-Like Z-Shaped Tunnel Field-Effect Transistor With Improved Ambipolar Behavior and RF Performance. IEEE Transactions on Electron Devices. 2017, 64(11), 4752-4758. DOI: 10.1109/TED.2017.2755507.
[19] Sachid, A. B.; Manoj, C. R.; Sharma, D. K.; Rao, V. R. Gate fringe-induced barrier lowering in underlap FinFET structures and its optimization. IEEE Electron Device Letters. 2008, 29(1), 128-130. DOI: 10.1109/led.2007.911974.
[20] Boucart, K.; Ionescu, A. M. Double-Gate Tunnel FET With High-k Gate Dielectric. IEEE Transactions on Electron Devices. 2007, 54(7), 1725-1733. DOI: 10.1109/TED.2007.899389.
[21] Abadi, R. M. I.; Saremi, M.A. Resonant Tunneling Nanowire Field Effect Transistor with Physical Contractions: A Negative Differential Resistance Device for Low Power Very Large Scale Integration Applications. Journal of Electronic Materials. 2018, 47(2), 1091-1098. DOI: 10.1007/s11664-017-5823-z.
[22] Gnani, E.; Gnudi, A.; Reggiani, S.; Baccarani, G. Theory of the Junctionless Nanowire FET. IEEE Transactions on Electron Devices. 2011, 58(9), 2903-2910. DOI: 10.1109/TED.2011.2159608.
Point 3: Why did the authors propose the z-gate layout just for NMOSs? How is the case for PMOSs?
Response 3: Thank you for your suggestion. The Z-gate layout for NMOSs is proposed for simplicity. The Z-gate layout for PMOSs is similar to that for NMOSs, except the doping of source/drain region is p-type and the substrate doping is n-type.
Point 4: How are the issues of the proposed layout in terms of fabrication and process aspects?
Response 4: Thank you for your suggestion.
In this paper, the proposed Z gate layout devices achieve total-dose hardness by eliminating these edges, but at the expense of fabrication feasibility due to the asymmetric active area design, as shown in Fig. 1 (c).
Point 5: It is better to combine Figs. 2-4 as Figs. 2(a), (b), and (c).
Response 5: Thank you for your suggestion. The manuscript has been modified based on your advice.
Figure 2. Simulation results of the Id-Vg curve of (a) single gate layout, (b) enclosed gate layout, (c) Z gate layout
Point 6: Did the authors propose the z-gate layout for the first time? If so, they should highlight this point.
Response 6: Thank you for your suggestion. The manuscript has been modified based on your advice.
In this paper, for the first time, a novel n-MOSFET layout with the Z gate with improved total ionizing dose (TID) tolerance is proposed.
In order to eliminate the parasitic path and overcome the disadvantages of an enclosed gate layout, for the first time, a novel n-MOSFET layout with the Z gate is proposed.
Point 7: Based on the presented results, the z-gate layout is not as good as the enclosed gate layout. Why should someone pick this structure for radiation-hardened applications?
Response 7: Thank you for your suggestion.
A widely studied layout with the radiation hardness called the enclosed gate layout [12-14], which requires the tradeoffs in application [14-15], is presented in Fig. 1(b). For instance, a very small width over length ratio (W/L) is not realistic for the Enclosed Layout Transistor (ELT), the minimum achievable W/L is 2.26 [15], which is a significant concern in analog circuits [16]. Moreover, a larger gate capacitance will cause a longer time delay which is not favorable for digital circuits. A large footprint is another disadvantage of the enclosed gate layout. In the circuit design, the area penalty induced by design has been the main drawback [17]. In order to eliminate the parasitic path and overcome the disadvantages of an enclosed gate layout, the novel n-MOSFET layout with the Z gate is proposed.
Point 8: The authors need to explain the models that they used in the 3D TCAD Sentaurus tool.
Response 8: Thank you for your suggestion. The manuscript has been modified based on your advice.
All simulations were performed using hydrodynamic model with high-field saturation and mobility degradation models including doping-dependence and carrier-carrier scattering.
Point 9: How is the z-gate layout compared with the enclosed gate layout when the channel length is scaled down to shorter than 120 nm?
Response 9: Thank you for your suggestion. When the channel length is scaled down to shorter than 120nm, all the devices suffer from short channel effects, and the Z-gate layout still effectively reduces the impact of the TID effect on the transistor performance compared with the single gate layout due to the parasitic path elimination of the layout.
Round 2
Reviewer 1 Report
The revision was made by properly taking into account the reviewers' comments. Thus, I recommend this paper can be accepted for publication.
Author Response
Response to Reviewer 1 Comments
Point 1: The revision was made by properly taking into account the reviewers' comments. Thus, I recommend this paper can be accepted for publication.
Response 1: Thank you for your suggestion.

Reviewer 2 Report
Dear Editor and Authors,
I have read the revision of the paper entitled "3-D Numerical Simulation of Z Gate Layout MOSFET for Radiation Tolerance". Although the Authors have made some changes, my feeling is that they are rather cosmetic, or in any case not as broad as a "MAJOR Revision" would have required. Specifically, I was concerned by the following aspects:
First, the Authors forgot or didn't even realize (!!??) that the comments of the Reviewer are neither made mainly for his benefit/information, nor for that of the Authors, but for that of the Readers. Therefore, all the elements highlighted by the Authors in their (rather short) reply should have been included in the revised paper. However, I did not see any of these included in the manuscript.
Second, the request to add new data related to transconductance, IDSS & IDSS/Ioff ratio have been glossed over and not even considered.
Third, the justification offered by the Authors for their 20 mV bias is shoddy at best. On one hand, they just cited another paper, which does not shed light on the very basic underlying reason why that paper did that in the first place, not to mention that the devices could be different in that paper compared to those presented in this paper. On the other hand, even the value itself still is not the same, as the work in the cited paper biased at 50 mV, not at 20 mV. Lastly, but not in the least, the main question related to the fact that this very low bias does NOT seem to correspond to real values used in practical ICs/circuits, has not even been addressed/answered.
Fourth, I still believe that at least SOME measured results should be included in the paper. I leave it to the latitude of the Editor to decide if this is an excessive request and can thus be discarded (at least from considerations of existing precedents, i.e. previous purely theoretical papers published in the same Journal).
Hence, I still maintain my previous recommendation that MAJOR corrections are needed.
With best wishes,
The reviewer
Author Response
Response to Reviewer Comments
Point 1: Since you discuss devices for improved radiation-hardened ICs, could it be that the ionization radiation may also have other effects, e.g. modify the number of surface states at the oxide-channel interface? Should these effects be considered in simulations as well, and if no, why? While you addressed this question to the reviewer, it is not very clearly explained in the manuscript, for the benefit of your future readers.
Response 1: Thank you for your suggestion. We neglected surface states created during irradiation at the oxide-channel interface for some reasons below.
When a CMOS device is exposed to radiation, hole trapping results in fixed charges and interface states in the thick oxides. According to a report on the radiation-induced fixed charge density and interface state density in MOS capacitors [1], the radiation-induced flat band voltage is dominantly shifted by the fixed charges. The effect of the interface states is minor. Therefore, in this simulation, the interface states were neglected and only the fixed charge density was modified to reflect the total ionizing dose effect [2].
The effects of interface traps are left out of the simulations due to a lack of empirical information about several parameters of interface traps such as trap energy, density, and capture cross section, which are necessary for accurate simulations [3].
In addition, we can see from the literature [3] that the tendencies of ΔVth and ΔSS extracted from the simulation results are in good agreement with those from the experimental data of 5 Mrad. Through the 3-D simulation results, they confirm that for sub-100 nm GAA MOSFETs, the fixed charges in the gate spacer dominantly determine ΔVth and ΔSS, i.e., the TID effect. Note that, interface traps are not taken in simulation in this paper. Although that may result in some disagreement in the current levels as obtained with the experimental counterparts, this case does not have much impact on our findings, because the focus of this paper is not on the exact values of currents but more on the general trends and relative results of Z gate, enclosed gate and single gate layouts due to TID effect.
[1] Fernandez-Martinez, P.; Cortes, I.; Hidalgo, S.; Flores, D. Palomo, F. R. Simulation of total ionizing dose in MOS capacitors. in Proc. 8th Electron Devices Conf. (in Spanish). 2011, DOI: 10.1109/sced.2011.5744251.
[2] Min, S. L.; Lee, H. C. Dummy Gate-Assisted n-MOSFET Layout for a Radiation-Tolerant Integrated Circuit. IEEE Transactions on Nuclear Science. 2013, 60(4), 3084-3091. DOI: 10.1109/tns.2013.2268390.
[3] Moon, J.-B.; Moon, D.-I.; Choi, Y.-K. Influence of Total Ionizing Dose on Sub-100 nm Gate-All-Around MOSFETs. IEEE Transactions on Nuclear Science. 2014, 61(3), 1420-1425. DOI: 10.1109/tns.2014.2319245.
Point 2: Add in the analysis/comparison, the data for transconductance, IDSS & IDSS/Ioff ratio or add a new small sections for each parameter. Otherwise, explain in the manuscript why you feel these to be unnecessary.
Response 2: Thank you for your suggestion. According to literature [2,4,5], the Id-Vg characteristics are simulated, and the data for leakage current increment and threshold-voltage shift is only extracted from the Id-Vg curves. As seen in literature [4], “The degradation of devices is mainly characterized by threshold-voltage shift and off-state leakage current.” Thus, in this paper, parameters of leakage current increment and threshold-voltage shift are fair enough to investigate the deteriorated radiation effects.
As we known, IDSS is the maximum current that flows through a FET transistor, which is when the gate voltage, VG, supplied to the FET is 0V, as shown in Fig.1. And it is only valid when the FET transistor is a JFET or depletion MOSFET. However, the proposed Z gate layout transistor is an enhanced MOSFET, we think the parameters of IDSS and IDSS/Ioff ratio are unnecessary to investigate.
Fig.1 Description of IDSS
[2] Min, S. L.; Lee, H. C. Dummy Gate-Assisted n-MOSFET Layout for a Radiation-Tolerant Integrated Circuit. IEEE Transactions on Nuclear Science. 2013, 60(4), 3084-3091. DOI: 10.1109/tns.2013.2268390.
[4] Wang, J.; Wang, W.; Huang, R.; Pei, Y.; Xue, S.; Wang, X. A.; Wang, Y. Deteriorated radiation effects impact on the characteristics of MOS transistors with multi-finger configuration. Microelectronics Reliability. 2010, 50(8), 1094-1097. DOI: 10.1016/j.microrel.2010.04.008.
[5] Turowski, M.; Raman, A.; Schrimpf, R. D. Nonuniform total-dose-induced charge distribution in shallow-trench isolation oxides. IEEE Transactions on Nuclear Science. 2004, 51(6), 3166-3171. DOI: 10.1109/tns.2004.839201.
Point 3: The reasons for using a 20mV drain bias is not fully clear. The explanation provided references to a publication which used 50mV for a different structure. Please either explain in the manuscript why 20mV is an appropriate choice or use a more realistic value. Simulating and comparing the performance at VDD drain bias would go a long way in addressing this concern. If 20mV simply gives the best results for the Z-gate layout in comparison to alternatives, then state this in the manuscript.
Response 3: Thank you for your suggestion. In this paper, the degradation of devices is mainly characterized by threshold-voltage shift and off-state leakage current. The threshold-voltage is determined by linear extrapolation method at VDS = 20 mV. This is one reason for using a 20mV drain bias. In addition, the use of 20mV drain bias gives the best results for the Z-gate layout in comparison to alternatives. We have stated this in the manuscript based on your advice.
The degradation of devices is mainly characterized by threshold-voltage shift and off-state leakage current. The threshold-voltage is determined by linear extrapolation method at VDS = 20 mV. Moreover, the use of 20 mV drain bias gives the best results for the Z-gate layout in comparison to the alternatives (have not shown here). Thus, in this paper the three layout types are simulated at the drain bias of 20 mV sweeping the gate bias from 0 V to 1.5 V.
Point 4: If possible, include some analysis on the effect of channel length scaling on radiation tolerance for the proposed layout.
Response 4: Thank you for your suggestion. The manuscript has been modified based on your advice.
In addition, the radiation effects will be even deteriorated with the channel length shrinking. This is a serious problem for highly scaled device operating under radiation environment.

Reviewer 3 Report
The revised manuscript has been improved but I still have few comments presented below:
1. The authors need to double-check the text and improve the writing level.
2. It will be good if the authors show a plot explaining the effect of channel length scaling on radiation tolerance for the proposed layout compared with the two others.
Totally, I think this current format of the manuscript does need Minor Revision.
Author Response
Response to Reviewer 3 Comments
Point 1: The authors need to double-check the text and improve the writing level.
Response 1: Thank you for your suggestion. The manuscript has been modified based on your advice.
Point 2: It will be good if the authors show a plot explaining the effect of channel length scaling on radiation tolerance for the proposed layout compared with the two others.
Response 2: Thank you for your suggestion. The manuscript has been modified based on your advice.
In addition, the radiation effects will be even deteriorated with the channel length shrinking. This is a serious problem for highly scaled device operating under radiation environment.
